# Reproducibility across single-cell RNA-seq protocols for spatial ordering analysis

**Morten Seirup**[1,2]*, **Li-Fang Chu**[2], **Srikumar Sengupta**[2], **Ning Leng**[2,3¤a], **Hadley Browder**[4], **Kevin Kapadia**[4], **Christina M. Shafer**[2], **Bret Duffin**[2], **Angela L. Elwell**[2¤b], **Jennifer M. Bolin**[2], **Scott Swanson**[2], **Ron Stewart**[2], **Christina Kendziorski**[3], **James A. Thomson**[2,5,6]*, **Rhonda Bacher**[7]*

**1** Molecular and Environmental Toxicology Program, University of Wisconsin Madison, Madison, Wisconsin, United States of America, **2** Morgridge Institute for Research, Madison, Wisconsin, United States of America, **3** Department of Biostatistics and Medical Informatics, University of Wisconsin Madison, Madison, Wisconsin, United States of America, **4** Department of Statistics, University of Florida, Gainesville, Florida, United States of America, **5** Department of Cell & Regenerative Biology, University of Wisconsin School of Medicine and Public Health, Madison, Wisconsin, United States of America, **6** Department of Molecular, Cellular, & Developmental Biology, University of California Santa Barbara, Santa Barbara, California, United States of America, **7** Department of Biostatistics, University of Florida, Gainesville, Florida, United States of America

¤a  Current address: Genentech, San Francisco, California, United States of America
¤b  Current address: Department of Genetics, University of North Carolina at Chapel Hill, Chapel Hill, North Carolina, United States of America
* seirup@wisc.edu (MT); jthomson@morgridgeinstitute.org (JAT); rbacher@ufl.edu (RB)

**Data Availability Statement:** The scRNA-seq data generated using the Smart-Seq protocol that support the findings of this study have been deposited in NCBI's Gene Expression Omnibus with the GEO Series accession code "GSE116140"

## Abstract

As newer single-cell protocols generate increasingly more cells at reduced sequencing depths, the value of a higher read depth may be overlooked. Using data from three different single-cell RNA-seq protocols that lend themselves to having either higher read depth (Smart-seq) or many cells (MARS-seq and 10X), we evaluate their ability to recapitulate biological signals in the context of spatial reconstruction. Overall, we find gene expression profiles after spatial reconstruction analysis are highly reproducible between datasets despite being generated by different protocols and using different computational algorithms. While UMI-based protocols such as 10X and MARS-seq allow for capturing more cells, Smart-seq's higher sensitivity and read-depth allow for analysis of lower expressed genes and isoforms. Additionally, we evaluate trade-offs for each protocol by performing subsampling analyses and find that optimizing the balance between sequencing depth and number of cells within a protocol is necessary for efficient use of resources. Our analysis emphasizes the importance of selecting a protocol based on the biological questions and features of interest.

## Introduction

Single-cell RNA sequencing (scRNA-seq) [1–5] is a powerful tool for studying transcriptional differences between individual cells. The innovation of droplet-based techniques [6, 7] and unique molecular identifiers (UMI) [8] has lowered the cost per cell and pushed the field

https://www.ncbi.nlm.nih.gov/geo/query/acc.cgi?acc=GSE116140. The 10X dataset from the Tabula Muris compendium public resource may be downloaded from https://figshare.com/articles/Robject_files_for_tissues_processed_by_Seurat/5821263. The MARS-seq dataset from Halpern et al. 2017 may be downloaded from GEO Series accession code " GSE84498" https://www.ncbi.nlm.nih.gov/geo/query/acc.cgi?acc=GSE84498. All code used in the analysis and figures is available on Github at https://github.com/rhondabacher/scSpatialReconstructCompare-Paper.

**Funding:** This publication was developed under Assistant Agreement number 83573701 awarded by the US Environmental Protection Agency to the University of Wisconsin Madison. It has not been formally reviewed by the EPA. The views expressed in this document are solely those of the authors and do not necessarily reflect those of the Agency. EPA does not endorse any products or commercial services mentioned in this publication. Funding for this research was also provided by the U.S. National Institutes of Health grant GM102756 (to C. K.). Genentech provided support in the form of salary for author N.L. No funders had any role in the study design, data collection and analysis, decision to publish, or preparation of the manuscript. The specific roles of these authors are articulated in the 'author contributions' section."

**Competing interests:** I have read the journal's policy and the authors of this manuscript have the following competing interests: N.L. is currently employed by Genentech. This does not alter our adherence to PLOS ONE policies on sharing data and materials.

toward obtaining data from tens of thousands of cells per experiment albeit at a reduced sequencing depth. Recent publications have compared the sensitivity, accuracy, and precision between several scRNA-seq techniques and report that the major trade-off between protocols is sensitivity, which is dependent on read depth [9, 10]. With the push for sequencing an ever-increasing number of cells at the expense of read depth per cell, the value of a higher read depth might be overlooked. Here we investigate the reproducibility of biological signals across protocols that naturally lend themselves to generating data on more cells versus higher read depth.

Studies comparing protocols have mainly done so with respect to performance on spike-ins or on technical variability alone [9, 10]. Recently, Guo et al. [11] showed agreement of cell types and signature genes between two platforms used for scRNA-seq–Fluidigm C1 and Drop-seq. However, few studies have examined comparative agreement among protocols for biological inferences beyond clustering and identifying differential gene expression, yet a key question of interest with single-cell data is its ability to reflect temporal or spatial heterogeneity. For cells collected at a given time, the underlying dynamic biological process is reflected in genome-wide differences in gene expression. Computational algorithms that attempt to order cells in time or space based on variability in gene expression have been developed [4, 12, 13], and more than 45 existing algorithms were recently compared [14]. Yet, as far as we know, no comparison of single-cell protocols exists for the question of cell ordering.

Here, our evaluation is in the context of spatial reconstruction in which we compared three independently produced scRNA-seq datasets on the mouse liver lobule. We chose to compare protocols on their ability to reflect the spatial patterning of the liver lobule in which the parenchymal cells of the liver, hepatocytes, are organized spatially in a polygonal shape around a central vein (Fig 1A). From the central vein, a gradient of metabolic functions is performed, extending to a portal triad at each vertex [15–19]. The gradient of differences in gene expression patterns is referred to as the zonation axis (from the central pericentral region outwards to the periportal region) [20]. This coordinated spatial organization provides a particularly interesting application of single-cell techniques. For this study, we obtained one dataset using Smart-seq—a full-length protocol, a second dataset using MARS-seq [21]—a UMI and plate-based protocol, and the third dataset generated using 10X [22]—a UMI and droplet-based protocol. Although the cell number and read depth differ greatly across datasets, we find high reproducibility of gene expression profiles after spatial reconstruction analysis. Given the reproducibility and that each protocol naturally lends itself to either producing more cells at a lower sequencing depth or fewer cells at a higher depth, our results demonstrate the importance of carefully evaluating the biological question and features of interest when selecting the appropriate sequencing protocol. In applications focused on lower expressed genes or on genes with high sequence similarity, increased read depth is preferable, whereas a focus on identifying cell types based on more highly expressed genes will benefit from collecting more cells. In an ideal situation a single cell assay would result in thousands of cells that are all sequenced at a high read depth, but technical and financial restrictions rarely make this possible.

## Results

### Differences in detection rates

By using the Fluidigm C1 system coupled with the Smart-seq protocol, we were able to identify on average around 38% (about 7,100 genes) of all genes in the genome expressed per cell, whereas the MARS-seq dataset finds on average 12% (about 2,200 genes) and the 10X dataset finds on average 6% (about 1,100 genes) (Fig 1B). This is in accordance with findings by

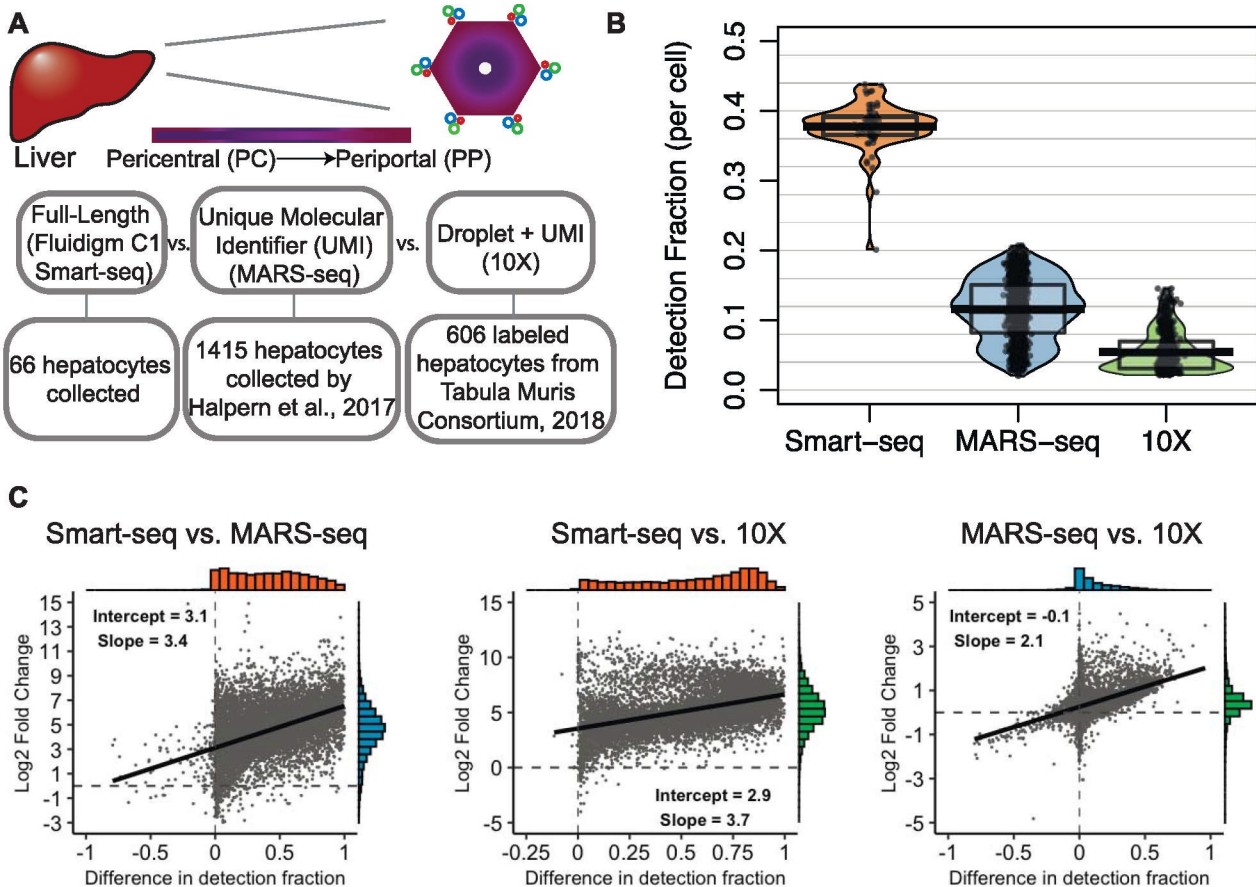

**Fig 1. Illustration of the liver anatomy and general comparison of the datasets.** A) Top. An illustration of the liver lobule identifying the portal triad along the outer edges and the central vein in the middle. The color gradient represents metabolic zonation. Bottom. A highlight of the main differences between the three datasets. B) Comparison of gene detection fraction between the datasets. The detection fraction per cell (y-axis) is shown for the three datasets (x-axis). C) Left. The log2 fold-change of genes detected above an average expression level of zero in the Smart-seq dataset compared to the MARS-seq dataset (y-axis) versus the difference in gene-level detection fractions between datasets (x-axis). A linear regression line is overlaid and histograms of the x- and y-axes are shown opposite of each axis. Middle. Similar plot shown for Smart-seq versus 10X. Right. Similar plot shown for MARS-seq versus 10X.

Ziegenhain et al. [9] when they examined various single-cell transcriptomic methods and by Phipson et al. [23] when they compared biases in full-length versus UMI protocols. The increased sensitivity of the full-length protocol is further illustrated in Fig 1C, which on a per-gene level shows the difference in detection fraction compared to the log fold-change in mean expression between the protocols. A difference in detection fraction of zero means that the gene is detected in the same fraction of cells in both datasets. The difference across protocols in log2 fold-change has a linear relationship with the difference in detection fractions, which indicates a fairly constant increase in log2 expression as cells are sequenced with greater sensitivity. At the intercept, a difference in detection equal to zero, the log2 fold-change is 3.1 between Smart-seq and MARS-seq, indicating an experiment wide increase in sensitivity in the Smart-seq protocol of approximately 9-fold. Between Smart-seq and 10X, the increase in sensitivity is approximately 12-fold, and there is a similar level of sensitivity between MARS-seq and 10X. Not surprisingly, the vast majority of genes are detected in a larger fraction of cells and have a higher expression level in the more deeply sequenced dataset using the Smart-seq protocol. However, it is worth pointing out that around 6% of genes have higher detection

using the MARS-seq protocol (negative values on x-axis), and a few of these genes also have higher expression levels (negative values on y-axis) than in the Smart-seq protocol. This subset of genes better detected in the MARS-seq dataset have higher GC content and are slightly longer (S1 Fig), which is consistent with previous reports of protocol comparisons [23, 24].

### Reconstructing the spatial organization of the liver lobule

Next, we reconstructed the spatial organization across the liver lobule by computationally ordering the cells in the three datasets according to their expression profiles. The MARS-seq dataset was spatially ordered by Halpern et al. [21] by first performing smFISH for six marker genes at various locations across the zonation axis, and dividing the pericentral to periportal axis into nine distinct zonation groups. Cells from scRNA-seq data obtained via MARS-seq were assigned to one of the nine groups based on each cell's expression profile of the six marker genes [21]. We ordered the cells in the 10X dataset using the Monocle2 algorithm, which builds an ordering of cells based on the expression similarity among the most highly variable genes [12]. For the Smart-seq protocol, we used the computational algorithm Wave-Crest to spatially order cells based on 15 marker genes known in the literature to be differentially expressed along the zonation axis (Fig 2A) [5]. All orderings assume the zonation profile and spatial organization can be represented in a single dimension. A similar reconstructed order was obtained for the Smart-seq dataset when applying Monocle2 (S2 Fig).

We next explored the dynamics of gene expression across the reconstructed pericentral to periportal axis. Since the MARS-seq dataset placed cells into nine discrete zones along the axis, we also divided the ordered cells from the Smart-seq and 10X datasets into nine equally sized groups in order to directly compare the gene expression profiles. The initial zonation groups are aligned with the pericentral region and the later zonation groups align with the periportal region. We also scaled gene expression as the dynamic range of counts differs greatly between datasets (Methods). The zonation profiles in Fig 2B of genes that are predicted to be highly regulated across the axis [21] have high agreement, with a median correlation of 0.95 between the three datasets. Before proceeding, we also performed an additional experiment to validate that our cell ordering and expression profiles reflect those of the liver lobule in vivo. Remarkably, immunohistochemistry studies showed that selected marker gene protein expression profiles also agreed with our spatial reconstructed scRNA-seq datasets: six markers display a pericentral-high/periportal-low profile and two markers display a pericentral-low/periportal-high profile in mouse liver lobule in vivo (Fig 2C). This confirmation in protein gradient patterns corresponding to our reconstructed spatial profiles provides us with confidence for further analysis on the biological inference in comparing the three protocols in this context.

### Comparing gene expression across the reconstructed liver zonation axes

An exciting prospect of single cell analysis is the identification of genes that have non-monotonic or dynamic expression across reconstructed time or space. Several genes in the bile acid synthesis pathway were shown by Halpern et al. [21] to be non-monotonically expressed in a pattern where the highest expression levels along the lobule correspond to the functional placement of the genes in the bile acid synthesis pathway (*Cyp7a1*, *Hsd3b7*, *Cyp8b1*, *Cyp27a1* and *Baat*) [21]. We find that the expression profiles for these genes are corroborated across the three datasets (Fig 3).

However, in the Smart-seq dataset, *Cyp8b1* is found to have flatter expression levels along most of the lobule and *Baat* appears to have an opposite trend in the 10X dataset. This variation may reflect differences in the datasets as both of these genes have been shown to be influenced by diet and circadian rhythms [25]. Other genes shown to be non-monotonically

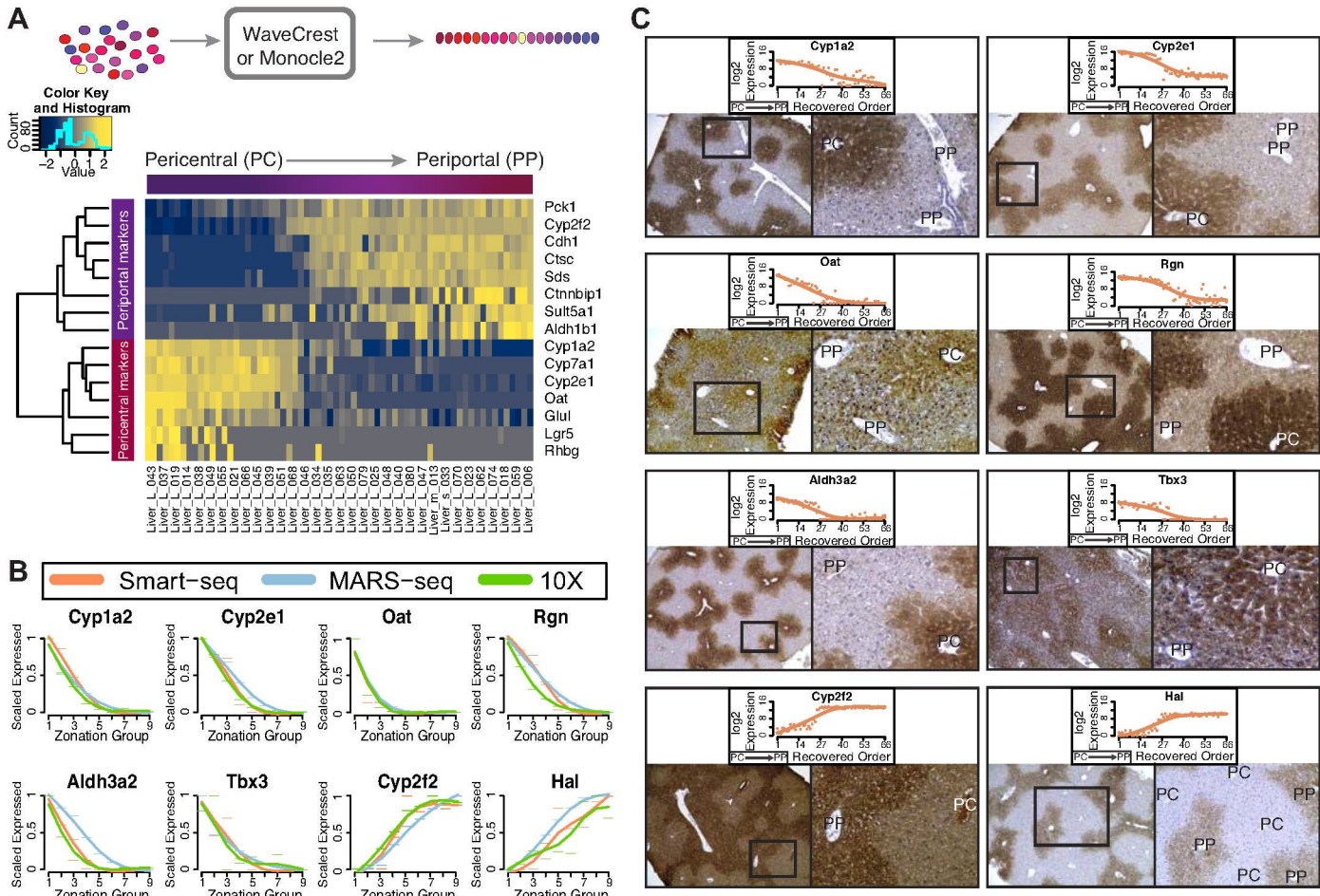

**Fig 2. Spatial ordering of hepatocytes and validation of dynamically expressed genes.** A) Top. Illustration of the spatial ordering process. Bottom. Heatmap showing the spatial ordering (x-axis) and the expression levels of the 15 marker genes (y-axis) for the Smart-seq dataset. Pericentral cells are found on the left-hand side, and periportal cells are found on the right-hand side. B) Scaled expression profile (y-axis) of eight dynamic genes based on the predicted spatial ordering (x-axis) of the Smart-seq dataset (orange), the MARS-seq dataset (blue), and the 10X dataset (green). C) Immunohistochemistry staining of the genes highlighted in B. Above the staining is the log2 expression counts (y-axis) across the reconstructed spatial order (x-axis) of the Smart-seq dataset. The left picture shows the staining and the right picture is an enlarged section (black square). PC = Pericentral, PP = Periportal.

expressed such as *Hamp*, *Igfbp2* and *Mup3* in Halpern et al. [21] display similar non-monotonic expression profiles in the Smart-seq and 10X datasets (Fig 3). We also investigated the expression pattern of *Glul* in more detail, as it is known to be expressed highly in a one hepatocyte-wide band around the central vein [26]. Accordingly, the predicted expression pattern found using all datasets demonstrated sufficient sampling of this region (S3 Fig). The ability to identify gene expression profiles that are either high at the pericentral end, high at the periportal end, or high in the middle of the liver lobule confirms that the sampling depth in all datasets is sufficient to spatially reconstruct the liver lobule.

We expanded our analysis to identify additional dynamic genes, those with significant differential expression along the reconstructed spatial order, by modeling gene expression as a function of the reconstructed zonation axis (Methods). Genes that were 'significantly zonated' in all datasets (adjusted p-value < .1) had highly correlated expression profiles. The Smart-seq versus MARS-seq expression profiles had the highest median correlation (0.86), while Smart-seq versus 10X had the lowest median correlation (0.69). The significantly zonated genes

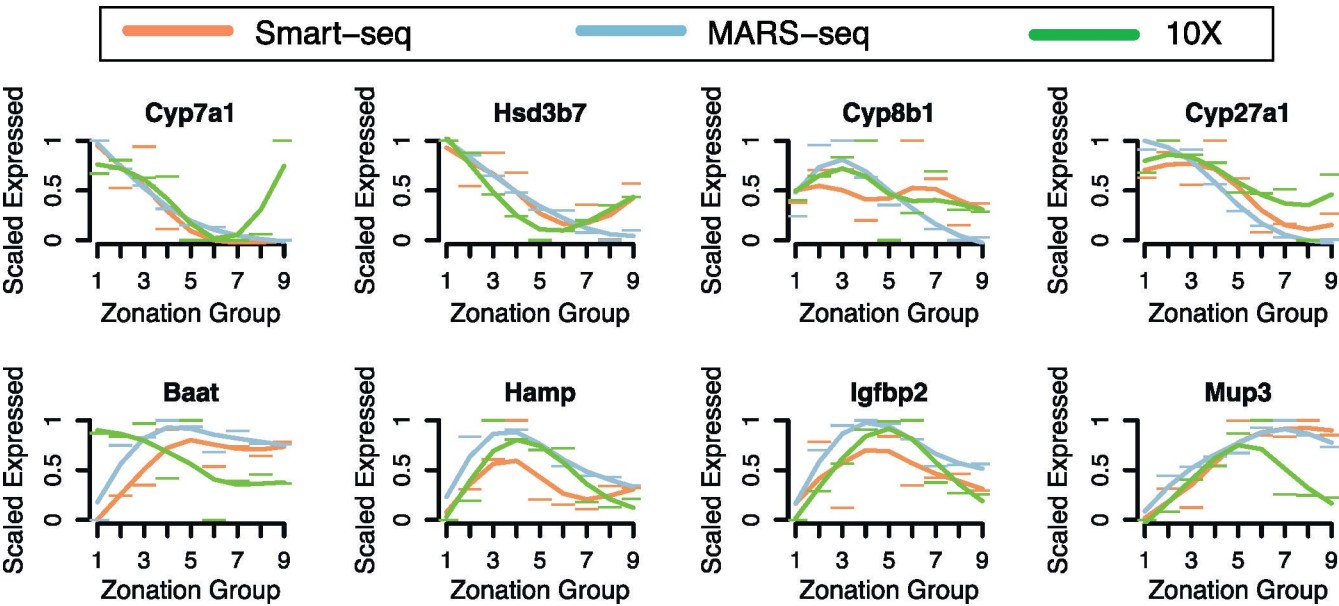

**Fig 3. Comparison of expression profiles across three datasets.** Scaled expression profile (y-axis) of eight genes non-monotonically expressed from Halpern et al. [21] along the spatial reordering (x-axis) of the Smart-seq dataset (orange), the MARS-seq dataset (blue), and the 10X dataset (green).

shared by all three datasets (Fig 4A) were enriched in KEGG metabolic processes with known periportal or pericentral bias such as amino acid metabolism (periportal), retinol metabolism (pericentral), and CYP450 metabolism (pericentral). Among the significantly zonated genes in the broad "Metabolic pathways" category in KEGG, the median correlation between all datasets ranged from 0.75 to 0.89 (Fig 4B). When all genes were considered the median correlation ranged from 0–0.04. A handful of genes were significantly zonated in all datasets but had low correlation in expression profiles (S4 Fig). We found these genes were generally influenced by

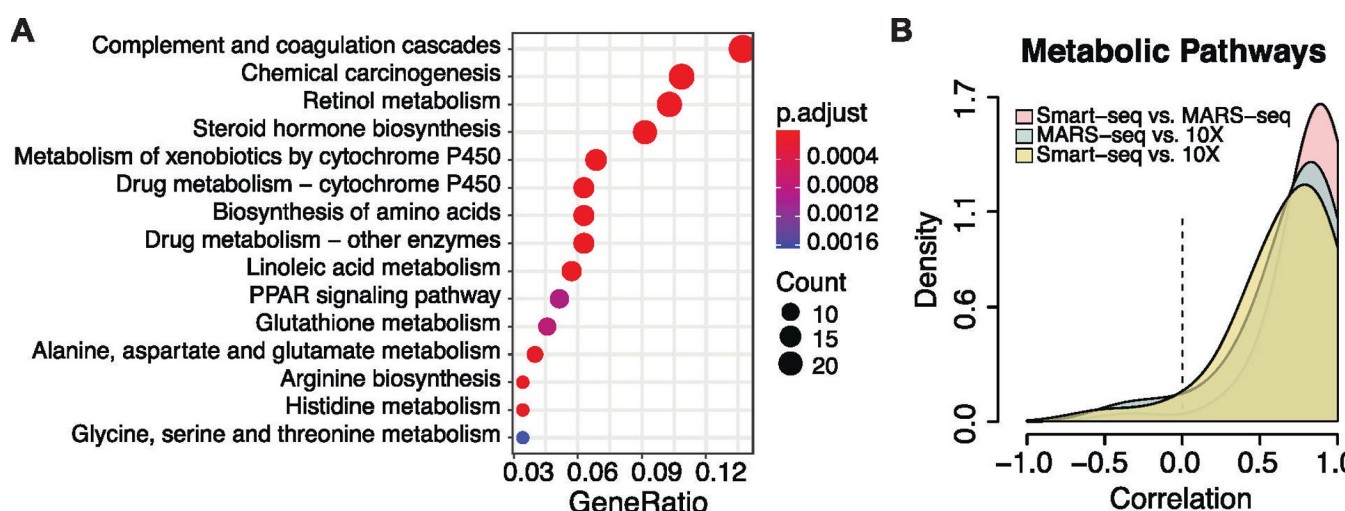

**Fig 4. Pathway analysis of significant genes and correlation of expression profiles.** A) KEGG enrichment analysis of genes with significant expression across the zonation groups in all three datasets. Dot size represents the fraction of enriched genes in each pathway, and the color represents the adjusted p-value for the enrichment. B) Correlation of significantly zonated genes in all three datasets annotated to the metabolic pathways in KEGG. The pairwise correlation is shown for each dataset comparison.

additional factors such as the circadian clock or diet (e.g. *Insig2* [27], *Mt1* and *Mt2* [28], *Scp2* [29], and *Mup2* [30]) or sex (e.g. *Apoc1* [31]). Despite using different reordering algorithms and protocols, the three datasets show high agreement of expression along the recovered peri-central to periportal axis among genes that are significantly zonated in all datasets, and reliably mirror the *in vivo* patterning of the liver lobule (S5 Fig).

## Differences in gene profiles among lowly expressed genes and gene isoforms

When we look at genes with moderate and low expression levels, we find that the datasets dif-fer to a greater degree. We identified 21 genes that were classified as significantly zonated along the pericentral to periportal axis in the Smart-seq dataset that were not detected at all in the MARS-seq dataset and 35 such genes not detected in the 10X dataset. Compared to the Smart-seq dataset, 10 genes were exclusively detected in the MARS-seq dataset and no genes were exclusive to the 10X dataset. Fig 5A shows the six most highly expressed genes that we were able to exclusively identify in the Smart-seq dataset having significant zonation (adjusted p-value < .1).

Beyond gene-level expression dynamics, we also evaluated isoform analysis–another excit-ing field of study with scRNA-seq data [32–34]. Many genes in the genome have two or more

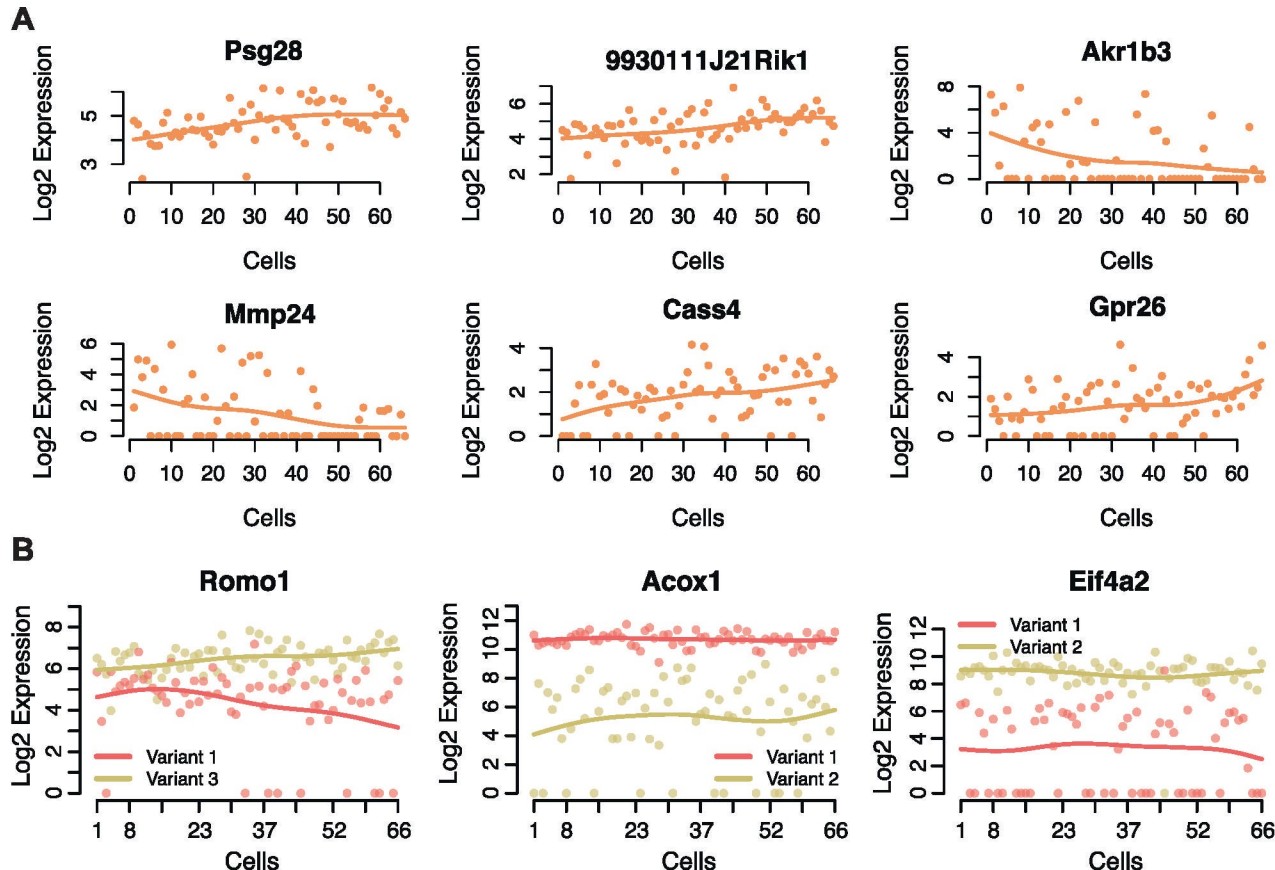

**Fig 5. Genes and isoforms found in the full-length dataset and not in the UMI datasets.** A) Six genes found to be significantly zonated in the Smart-seq dataset that were not detected in either the MARS-seq or 10X datasets. The log2 of expression values are represented on the y-axis and the spatially ordered cells are found on the x-axis. B) Examples of genes with two transcript variants expressed differently across reordered cells from the Smart-seq dataset.

isoforms that are distinctly expressed and can change properties such as structure, function, and localization of the resulting protein [35]. Due to the increased sensitivity of full-length cDNA libraries generated by Smart-seq protocol, we were able to examine genes with known isoforms and identify cases where the transcript variants for each isoform has distinct expression from each other across the pericentral to periportal axis, which is not possible with less sensitive protocols. In Fig 5B the transcript variants of *Romo1* are seen to display opposite trends in expression across the zonation axis, where the *Romo1* variant 3 is increasing in expression from the pericentral end toward the periportal end and the *Romo1* variant 1 is decreasing in expression along the same axis. We also highlight genes *Acox1* and *Eif4a2*, whose variants both show constant expression across the zonation axis but at different levels. Both of these genes are known to have isoform-specific expression in the liver lobule [36, 37] (for Ensembl and ENTEREZ IDs for transcript variants see S1 Table).

Due to UMI-based protocols capturing only one end of the transcript compared to full-length cDNA procedures, there is an inability to resolve not just isoforms but also many genes that are closely related. For instance, there were 242 concatenated genes in the MARS-seq set that correspond to 539 unique genes. An example of this is seen in S6 Fig where we highlight a concatenate of *Ugt1a* enzymes. Eight genes are concatenated (annotated together) and when combined, the average expression level is shown to be high at the pericentral end of the lobule and low at the periportal end in both the MARS-seq and Smart-seq datasets. Again, it is clear that not all the members of this concatenated group follow this trend as *Ugt1a6a* can be seen to have consistent expression levels across the pericentral to periportal axis in the Smart-seq data.

## Comparing the trade-offs of cell number versus sequencing depth within each protocol in silico

In a further comparison of the protocols we performed a subsampling experiment to study the trade-offs between higher depth versus more cells. For each dataset, we held either the number of cells or the sequencing depth constant while varying the other. For the Smart-seq and 10X datasets, we evaluated the effect on the cell ordering as well as the gene-specific expression profiles across the zonation axis. For the MARS-seq dataset, we only evaluated the effect on expression profiles as the assignment of each cell to a zonation group depended on external data and was independent of the other cells profiled. We estimated the mean squared error (MSE) as the difference in zonation profiles in the subsampled dataset versus the original dataset (Methods). In Fig 6A, the MARS-seq dataset displayed an approximately linear tradeoff in expression profile error for fewer cells at the original read depth. However, at reduced read depths using the original 1,415 cells, the error increased exponentially (Fig 6B). Within a dataset, we can compare the MSE between the two trade-off scenarios. For the MARS-seq dataset, resequencing at the same depth results in error that is equivalent to the reduction observed in MSE by going from 600 to 1,400 total cells. For the 10X dataset, we also find an approximately linear tradeoff in expression profile error for fewer cells at the original read depth (Fig 6C). However, at reduced read depths using the original 606 cells, we observe a gradual increase in error as total depth decreases (Fig 6D). Similarly, by comparing the MSE trade-off, it appears that resequencing at the same depth results in error that is equivalent to reducing the total cells from 600 to around 400. Thus, in scenarios with very low sequencing depth (average of 3–12k total UMIs per cell), sequencing deeper may be more beneficial than adding more cells. For the Smart-seq dataset, we found the spatial ordering to be quite robust to reduced sequencing depth, even as low as 50% fewer reads only marginally increased the average MSE (Fig 6F). The average sequencing depth for the

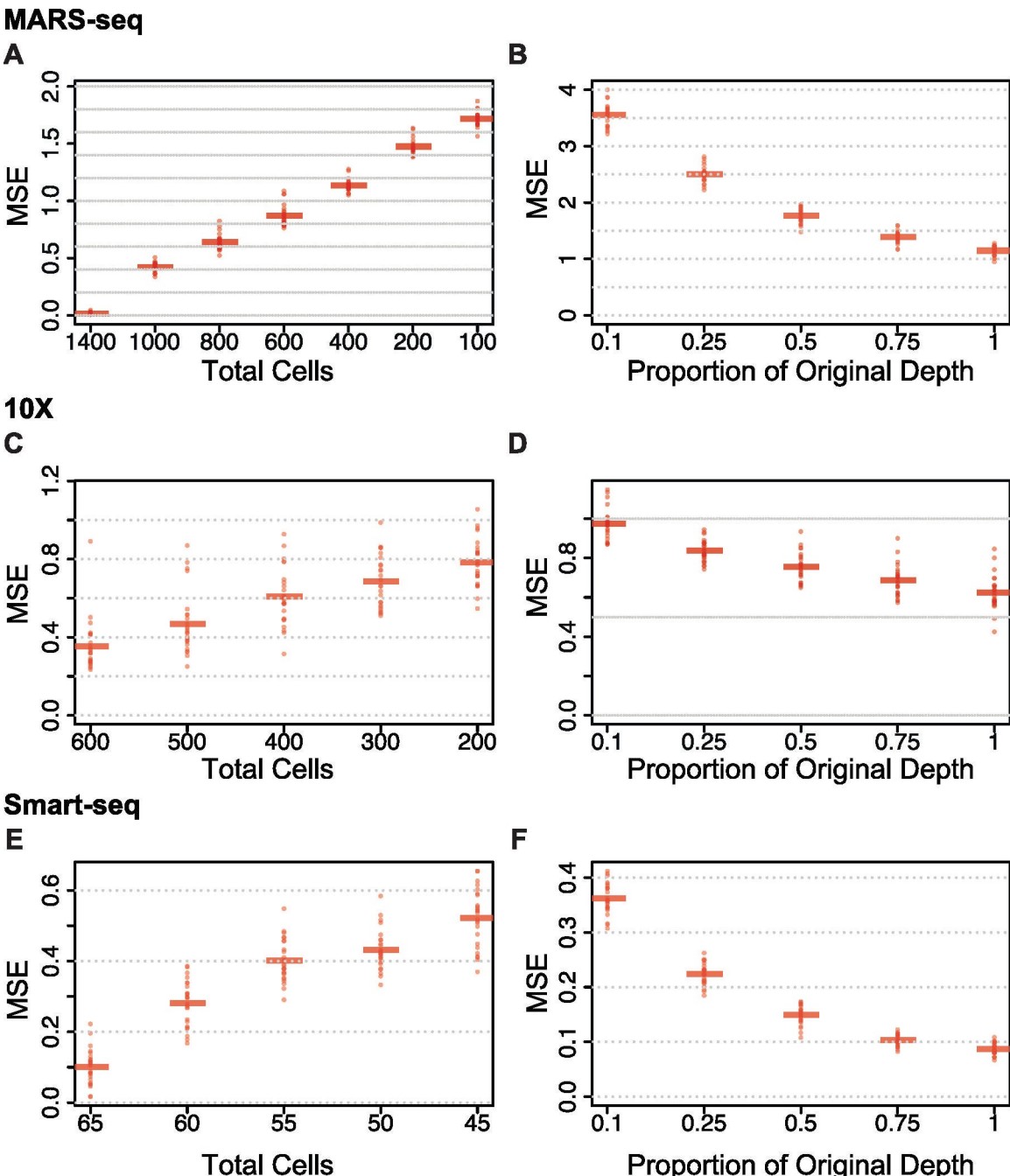

**Fig 6. Subsampling total numbers of cells and sequencing depth.** A) For 25 subsamplings at various total numbers of cells in the MARS-seq dataset, the MSE of the expression profile over 500 randomly selected genes is shown. B) Similar to A, but for 25 subsamplings at various total read depths. C&D) Similar to A&B, but for the Smart-seq dataset. E&F) Similar to A&B, but for the 10X dataset.

Smart-seq cells was 3.5 million counts per cell, well beyond the suggested sequencing saturation for single-cell data that has been estimated to occur close to one million total reads [38]. We do see more dramatic increases in error related to expression profiles when collecting fewer cells (Fig 6E). For Smart-seq data, sequencing to even half of the current depth and increasing the number of cells would be beneficial.

## Discussion

In summary, we compared three scRNA-seq datasets of mouse hepatocytes where two, MARS-seq and 10X, are wide but shallow and the other, Smart-seq, is narrow but deeply sequenced. We find that the three different protocols present highly reproducible liver zonation profiles in single cells, and for the vast majority of genes that are highly expressed we observe highly comparable results. Our results were not dependent on any one computational method or preprocessing pipeline.

We acknowledge that inferring spatial information in silico has limitations and requires extensive validation. Other methods are being developed that directly detect the spatial pattern of mRNA expression such as small molecule in-situ hybridization [39, 40] where mRNA molecules in tissue sections are directly labeled by complementary probes allowing visualization of the spatial location. Unfortunately, this approach is expensive, labor intensive, and low throughput. Another recently developed technique is spatial sequencing [41–44] where a tissue slice is placed on top of an array of pre-barcoded oligo(dT) primers and imaged. The mRNA is then bound to the primers in the array and the original spatial location of an mRNA is subsequently inferred based on the sequenced barcode. While this method provides transcriptomic data at an unprecedented accuracy and high spatial resolution, the cells are sequenced at lower total depths and is limited to the detection of the 3' end of the mRNA. The sampling technique is further prone to sample mRNA from more than one cell and thus lowers the transcriptomic resolution compared to single-cell sequencing [41].

We find that when we look at medium to low expressed genes, the increased sensitivity of the Smart-seq protocol is able to identify several exclusive genes, as well as, isoforms that behaved differently across the pericentral to periportal axis. Though in general there are still limitations of short reads in regard to isoform analysis and if more accuracy is needed, the newly developed technique ScISOr-seq [45] might be better suited. We do however believe that this full-length data allows for more reliable preliminary isoform analysis compared to either UMI method. Nevertheless, the main weakness of using fewer cells is that it is less likely that rare cell types will be sampled. In cases where such rare cells are of high interest, protocols that produce a large number of cells are preferable. In an ideal case, one would sample many cells and sequence all of them deeply; unfortunately, this is not always possible in practice and the decision of whether to sample many cells shallowly or fewer cells deeply comes down to whether rare cell types are of interest or if higher resolution of the individual cells is preferred.

Given the distinct advantages of the protocols, we emphasize that the biological question should be the driving factor when deciding on protocol. Within a chosen protocol, achieving balance between the sequencing depth and the number of cells is still an important consideration for optimal use of resources. Based on our simulations of datasets at opposite ends of the sequencing depth versus number of cells trade-off, there is eventually a detriment to sacrificing reads for additional cells or sequencing beyond the attainable sensitivity level on too few cells. We expect that the extent of the cells versus depth trade-off will vary for other cell types or tissues and it will largely depend on the heterogeneity of the biological system under study.

## Methods

### Animals and handling

All animals were kept under standard husbandry conditions. A wildtype 8-week-old male C57BL/6 (Jackson Laboratories) was used in this experiment. Using isoflurane, the mouse was anesthetized before euthanizing by cervical dislocation. Animal experiments and procedures were approved by the University of Wisconsin Medical School's Animal Care and Use

Committee and conducted in accordance with the Animal Welfare Act and Health Research Extension Act.

## Cell isolation

The euthanized mouse was pinned to a Styrofoam plate using 20 ga needles to aid in dissection. The abdominal cavity was opened, and the portal vein exposed. A piece of 4–0 suture thread (Ethicon vicryl coated) was threaded under the portal vein and used to secure a 26 ga catheter inserted into the portal vein (Butler Schein animal health 26 G IV Catheter, Fisher Scientific). Hepatocytes were isolated using a 2-step perfusion protocol. First, Liver Perfusion Medium (Gibco) warmed to 37˚C was pumped through the catheter for 10 minutes using a peristaltic pump at 7 ml/min flowrate. Then, Liver Digest Medium (Gibco) warmed to 37˚C was pumped through the liver at the same settings for 10 minutes. After perfusion, the liver was excised and transferred to a 10 cm dish containing 20 ml liver digest medium. The liver was dissected, allowing the cells to spill into the media. The cells were then filtered through a 40 μm cell strainer into a 50 ml tube and 30 ml media (Williams E media + 2 μg/ml human insulin + 1x glutamax + 10% FBS) were added and placed on ice. The hepatocytes were purified by centrifugation at 50 x G, 4 times for 3 minutes each, each time discarding the supernatant and adding media.

## Single-cell RNA sequencing: Smart-seq dataset

Single-cell RNA sequencing was performed as previously described [4, 5] with the following modifications. In this study, we used small (5–10 μm), medium (10–17 μm), and large (17–25 μm) plate sizes. ERCC RNA Spike-In (ThermoFisher Cat. No. 4456740) was diluted in the lysis mix following the manufacturer's user guide and previous studies [46]. Single end reads of 51 bp were sequenced on an Illumina HiSeq 2500 system. Sequencer outputs were processed using Illumina's CASAVA-1.8.2. The demultiplexed reads were trimmed and filtered to eliminate adapter sequence and low-quality basecalls. The reads were mapped to an mm10 mRNA transcript reference (extended with ERCC transcripts) using bowtie-0.12.9 [47]; expression estimates were generated using RSEM v.1.2.3. [48]. Using the Fluidigm C1 system to capture and synthesize cDNA from single cells in the liver, we generated transcriptomes for 149 cells. To exclude low quality transcriptomes, we removed cells in which the fraction of ERCC spike-in made up 20% or more of the total assigned reads. This left 66 high quality cells that were used in the downstream analysis. Finally, the data was normalized using SCnorm (R package v.1.5.7) [49].

## Spatial reordering: Smart-seq dataset

For the Smart-seq data, the cells were computationally ordered using the Wave-Crest method as described in Chu et al. [5]. For the reordering step, gene expression values were rescaled to mean 0 and variance 1 to ensure the values across different genes are comparable. The WaveCrest algorithm implements an extended nearest insertion algorithm that iteratively adds cells to the order and selects the insertion location as the location producing the smallest mean squared error in a linear regression of the proposed order versus gene expression. A 2-opt algorithm is then used to find an optimal cell order by considering adjacent cell exchanges. The cell ordering step uses the expression profiles of preselected known marker genes of liver zonation. Thus, the resulting linear profile of ordered cells represents the pericentral to periportal axis. The known marker genes used to construct the pericentral to periportal axis in Wave-Crest include the following pericentral markers: cytochrome P450 7a1 (*Cyp7a1*), cytochrome P450 2e1 (*Cyp2e1*), ornithine aminotransferase (*Oat*), cytochrome P450

1a2 (*Cyp1a2*), rh family, B glycoprotein (*Rhbg*), leucine-rich repeat-containing G-protein coupled receptor 5 (*Lgr5*), glutamate-ammonia ligase (*Glul*); and the following periportal markers: phosphoenolpyruvate carboxykinase 1 (*Pck1*), catenin beta interacting protein 1 (*Ctnnbip1*), aldehyde dehydrogenase 1 family member B1 (*Aldh1b1*), sulfotransferase family 5A, member 1 (*Sult5a1*), cytochrome P450 2f2 (*Cyp2f2*), cathepsin C (*Ctsc*), serine dehydratase (*Sds*), and E-cadherin (*Cdh1*). All markers were selected based on their expression ratio as reported by Braeuning et al. [20].

A detection step was done to identify additional genes that are dynamic across the one-dimensional pericentral to periportal axis by fitting a linear regression to the relationship between each gene's expression as a function of the Wave-Crest cell order. To determine if a gene is significantly dynamic (differentially zonated) along the recovered axis, we tested whether the regression slope is different from zero. We reported the Benjamini-Hochberg adjusted p-values to control the false discovery rate. For genes having an adjusted p-value < .01, the direction of the expression profile was assigned based on the sign of the regression slope (periportal: positive slope, pericentral: negative slope). We also calculated the linear fitting MSE for each gene. Genes with a smoother trend over the recovered cell order are expected to have a smaller MSE. We report the full list of genes, sorted by their MSE, in S2 Table; scatter plots for genes having adjusted p-value < .01 are shown in S1 File; and the normalized gene expression matrix with cells in the WaveCrest order is in S1 Dataset.

## Spatial reordering: 10X dataset

The 10X dataset was downloaded from the Tabula Muris compendium public resource via figshare [22]. The 10X data was originally processed using the CellRanger v.2.0.1. Within the liver cells, the authors originally identified 975 hepatocytes. For our analysis, we performed a second quality control step to remove cells with low RNA content, possible doublets, or dead/damaged cells, where we filtered cells based on the total number of genes expressed per cell. Using the Seurat R package v.3.1.5, hepatocytes were further filtered to those having between 200 and 3,000 genes detected per cell (only one cell had more than 5,000 genes detected per cell). Next, we clustered the cells using Seurat, where a k-nearest neighbors (KNN) graph was constructed based on the first 20 principle components to create a shared nearest neighbors graph based on the Jaccard index between each cell and its 20 nearest neighbors, as implemented in the FindNeighbors function. Clusters were then identified by partitioning this graph using the Louvain community detection algorithm with a resolution of 0.5, as implemented in the FindClusters function. The cells clustered into three distinct larger groups, and we retained only the largest grouping of cells that clustered together, resulting in 606 total cells. The data was then normalized using scran v.1.12.1 [50]. Next, we used Monocle v.2.12.0 [51] to order the cells, basing the ordering on the top 200 highly variable genes estimated using the mean variance relationship via the FindVariableFeatures function in Seurat. To determine if a gene is significantly dynamic (differentially zonated) along the recovered axis, the Monocle2 function differentialGeneTest was used to fit a spline on gene expression versus the estimated spatial order.

## Comparative analysis

Smoothed densities (bean plots) with overlaid raw data, the mean, and a box representing the interquartile range of the cellular detection fractions were created using the pirateplot function in the yarrr R package v.0.1.5. The cellular detection fraction was calculated per cell as the proportion of genes having expression greater than zero. The fold-change for each gene between two datasets (A versus B) was calculated as the log2 fold-change of dataset A over dataset B,

where each gene mean was calculated as the average expression among non-zero counts across all cells in the datasets. The heatmap in Fig 2 of marker gene expression on the normalized Smart-seq data was generated by setting values above the 95th percentile or below the 5th percentile to the 95th percentile or 5th percentile value, respectively.

Due to the datasets having different dynamic ranges, we used scaled expression plots to compare expression profiles. First the ordered cells in the Smart-seq dataset and 10X were each divided into nine equally sized groups to correspond to the nine zonation groups in the MARS-seq dataset. For a given gene, the median (Smart-seq) or mean (10X) expression in each group was calculated, then the nine values were scaled between zero and one. The significantly zonated genes for the Smart-seq and 10X datasets were identified as described previously. Genes were identified as significantly zonated in the MARS-seq dataset in Halpern et al. [21] using the Kruskal-Wallis nonparametric test on the mean expression profiles across the zonation groups for each gene. The p-value threshold for the significantly zonated designation was set to .10.

In the zonation group scatter plots, the mean or median expression is shown for each group and smoothed fits were overlaid using the smooth.spline function in R with the degrees of freedom parameter df = 4. Expression correlations along the zonation axis between datasets were calculated using Pearson correlation. Enrichment of genes in KEGG pathways was done using the R package clusterProfiler (v.3.10.1) [52]. For the enrichment analysis, since different statistical methods were used to assess zonation profiles, genes were considered significantly zonated if they had an adjusted p-value < .1 in all datasets. Additional KEGG categories from this analysis can be interactively viewed on GitHub https://github.com/rhondabacher/scSpatialReconstructCompare-Paper.

## Subsampling analysis

In all subsamplings described below, each scenario was repeated a total of 25 times and the zonation group means were scaled to be between zero and one.

For the MARS-seq dataset, zonation group means were recalculated on a subsampled set of cells using the posterior probability matrix and original UMI counts from Halpern et al. [21]. In each sampling, the MSE was calculated based on a random sample of 500 genes as $\sum_{i=1}^{500} \sum_{j=1}^{9} \left( Z_{i,j} - \hat{Z}_{i,j} \right)^2 / 500$, where $Z_{i,j}$ represents the mean expression of gene $i$ in zonation group $j$ in the original dataset and $\hat{Z}_{i,j}$ is the corresponding value for the subsampled dataset. For subsampling at lower read depths, we fixed the number of cells at the original total of 1,415 cells and simulated each cell's gene counts individually using a multinomial distribution. For each cell the parameters of the multinomial distribution were estimated as: the subsampled total counts were set to X% of the original total read counts for that cell (for X = (10,25,50,75,100)) and each gene's cell-specific probability was calculated as its original count divided by the original total counts for that cell. The MSE was calculated for each subsampled set as previously described.

For the Smart-seq dataset, we reran WaveCrest when subsampling the total number of cells using the original parameter settings and marker genes. Then, as before, the ordered cells were assigned zonation groups by dividing cells into nine equally sized groups. The zonation profile error was estimated using MSE and calculated as previously described, with the exception that since Wave-Crest orders can be flipped, we calculated the MSE on the returned order and its reverse and kept the minimum MSE of the two. To evaluate the expression profile error at lower read depths, we used a similar approach as described earlier for the MARS-seq dataset, fixing the number of cells to be the same as the original total of 66 and, since the order correlation was shown to be consistently high, we used the original WaveCrest order for every

scenario when evaluating zonation profile error. For the 10X dataset, the subsampling was performed similarly as for the Smart-seq dataset, however Monocle2's ordering was more variable as it was not based on marker genes, and thus we did not fix the order when evaluating the expression profile error. Trade-offs in MSE are directly comparable within a dataset but due to intrinsic differences in the original processing and in subsampling, the MSE should not be compared across the datasets.

## Immunohistochemistry

An 8-week-old male C57BL/6 mouse was anesthetized using isoflurane before euthanizing by cervical dislocation. The liver was excised, sliced as thinly as possible with a razor blade, and fixed in formaldehyde overnight. The liver slices were paraffin embedded and sectioned. Sections were stained following the protocol published by Abcam (http://www.abcam.com/ps/pdf/protocols/ihc_p.pdf). In short, the slices are deparaffinized by dipping into sequential solutions of 100% xylene, 50–50% xylene-ethanol, 100% ethanol, 95% ethanol, 70% ethanol, 50% ethanol, and tap water. The antigens were then retrieved by placing the slides in Tris-EDTA buffer (10 mM Tris Base, 1 mM EDTA Solution, 0.05% Tween 20, pH 9.0) and incubating them in a decloaking chamber (Biocare Medical Decloaking Chamber #DC2008US) with the following settings: delayed start 30 sec.; preheat 80˚C, 2 min.; heat 101˚C, 3 min. 30 sec.; and fan on. The slides were washed 2 x 5 min in TBS + 0.025% Triton X-100 before they were blocked for two hours at room temperature in 10% normal serum in 1% BSA. The appropriate primary antibody was then diluted in the same 10% normal serum in 1% BSA, added to the slides, and incubated at 4˚C overnight in an incubation chamber. The next day the slides were washed 2 x 5 min in TBS + 0.025% Triton X-100 followed by 15 min incubation in 0.3% $H_2O_2$ at room temperature. Next, the appropriate secondary antibody was diluted into 10% normal serum in 1% BSA before it was added to the slides and incubated for 1 hour at room temperature. The slides were then washed 3 x 5 min in TBS before DAB (#ab103723) staining mixed according to manufacturer instruction was applied and incubated under a microscope to stop the reaction after sufficient staining. The slides were rinsed in tap water for 5 min before being counterstained with Mayer's hematoxylin (#MHS1-100ML) for 30 sec. The stain was developed in running tap water for 5 min. The slides were then dehydrated by sequentially dipping in 50% ethanol, 70% ethanol, 95% ethanol, 100% ethanol, 50–50% xylene-ethanol, and 100% xylene before Poly-Mount (#08381–120) was added and a coverslip placed on top. The following primary antibodies were added: Aldh3a4 1:250 (AB184171), Cyp2e1 1:50 (AB28146), Cyp1a2 1:50 (R31007), Rgn 1:100 (NBP1-80849), Oat 1:50 (AB137679), Cyp2f2 1:100 (SC-67283), Hal 1:50 (AV45694), and Tbx3 1:50 (SC-31657). The following secondary antibodies were used: goat-anti-rabbit HRP conjugated (ab97051) and donkey-anti-goat HRP conjugated (ab97110) at a concentration of 1:500.

## Supporting information

**S1 Fig. Examining GC content and gene length biases in detection rates across datasets.** Top) The GC content (left) and gene length (right) are shown for genes having a higher detection fraction in either the Smart-seq dataset (gray) or the MARS-seq dataset (blue). A dotted line is shown for genes having a larger mean in either dataset. The two lines closely correspond since the genes having a high detection fraction typically have a higher mean. Bottom) Similar to the top for comparing the Smart-seq and 10X datasets.
(PDF)

**S2 Fig. Correlation between WaveCrest and Monocle2 algorithms for ordering cells in the Smart-seq dataset.**
(PDF)

**S3 Fig. Expression of Glul.** Scaled expression plots of Glul showing high correlation among all three datasets.
(PDF)

**S4 Fig. Significantly zonated genes with low correlations.** Top left: Pairwise correlations of the expression profiles of all significantly zonated genes. Scatter plots are shown for all genes with correlation less than zero in at least one pairwise correlation.
(PDF)

**S5 Fig. Correlation analysis of specific KEGG pathways.** A) Top left: Correlation analysis for genes in the KEGG pathway "Complement and coagulation cascade". The pairwise correlations are shown for each dataset comparison. Following are plots for the five highest correlated genes in that pathway. B) Similar to (A) but for the "Retinol metabolism" pathway. C) Similar to (A) but for the "Drug metabolism–cytochrome P450" pathway.
(PDF)

**S6 Fig. Additional genes in Smart-seq dataset but not in the MARS-seq dataset.** Eight Ugt1a genes that were concatenated in the MARS-seq dataset (blue on all graphs), but can be resolved in the Smart-seq dataset (orange line).
(PDF)

**S1 Table. Ensembl and RefSeq IDs for genes with transcript variants.**
(XLSX)

**S2 Table. Summary of genes with dynamic expression across the zonation axis identified using WaveCrest.**
(CSV)

**S1 File. Scatter plots of dynamic genes listed in S2 Table.**
(PDF)

**S1 Dataset. Normalized Smart-Seq single-cell data with cells in the WaveCrest order.**
(CSV)

## Author Contributions

**Conceptualization:** Morten Seirup, James A. Thomson, Rhonda Bacher.

**Data curation:** Morten Seirup, Christina M. Shafer, Scott Swanson, Rhonda Bacher.

**Formal analysis:** Ning Leng, Rhonda Bacher.

**Funding acquisition:** James A. Thomson.

**Investigation:** Morten Seirup, Li-Fang Chu, Srikumar Sengupta, Bret Duffin, Angela L. Elwell, Jennifer M. Bolin.

**Methodology:** Morten Seirup.

**Project administration:** Morten Seirup, James A. Thomson.

**Resources:** James A. Thomson.

**Software:** Hadley Browder, Kevin Kapadia, Scott Swanson, Rhonda Bacher.

**Supervision:** Morten Seirup, Ron Stewart, Christina Kendziorski, Rhonda Bacher.

**Validation:** Morten Seirup, Scott Swanson.

**Visualization:** Hadley Browder, Kevin Kapadia, Rhonda Bacher.

**Writing – original draft:** Morten Seirup, James A. Thomson, Rhonda Bacher.

**Writing – review & editing:** Morten Seirup, Li-Fang Chu, Srikumar Sengupta, Ning Leng, Hadley Browder, Kevin Kapadia, Christina M. Shafer, Bret Duffin, Angela L. Elwell, Jennifer M. Bolin, Scott Swanson, Ron Stewart, Christina Kendziorski, James A. Thomson, Rhonda Bacher.

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
