## [Decision Letter · Decision Letter 0]

6 Jul 2020

PONE-D-20-17601

Reproducibility across single-cell RNA-seq protocols for spatial ordering analysis

PLOS ONE

Dear Dr. Bacher,

Thank you for submitting your manuscript to PLOS ONE. After careful consideration, we feel that it has merit but does not fully meet PLOS ONE’s publication criteria as it currently stands. Therefore, we invite you to submit a revised version of the manuscript that addresses the points raised during the review process.

One reviewer is supportive to the manuscript. However, another reviewer raised some concern. Please carefully address these comments. 

We look forward to receiving your revised manuscript.

Kind regards,

Zhong-Hua Chen, Ph.D.

Academic Editor

PLOS ONE

Journal Requirements:

We note that one or more of the authors are employed by a commercial company: Genentech.

2.1. Please provide an amended Funding Statement declaring this commercial affiliation, as well as a statement regarding the Role of Funders in your study. If the funding organization did not play a role in the study design, data collection and analysis, decision to publish, or preparation of the manuscript and only provided financial support in the form of authors' salaries and/or research materials, please review your statements relating to the author contributions, and ensure you have specifically and accurately indicated the role(s) that these authors had in your study. You can update author roles in the Author Contributions section of the online submission form.

3. We noted in your submission details that a portion of your manuscript may have been presented or published elsewhere.

"Two of the datasets that we analyze in our comparison are public and have been published (MARS-seq by Halpern et al., 2017 and the 10X by the Tabula Muris Consortium, 2018). This is noted throughout and properly cited. Note that we have submitted this manuscript as a pre-print to bioRxiv."

Please clarify whether this publication was peer-reviewed and formally published. If this work was previously peer-reviewed and published, in the cover letter please provide the reason that this work does not constitute dual publication and should be included in the current manuscript.

Reviewers' comments:

Reviewer's Responses to Questions

**Comments to the Author**

1. Is the manuscript technically sound, and do the data support the conclusions?

Reviewer #1: Partly

Reviewer #2: Yes

2. Has the statistical analysis been performed appropriately and rigorously? 

Reviewer #1: No

Reviewer #2: Yes

3. Have the authors made all data underlying the findings in their manuscript fully available?

Reviewer #1: Yes

Reviewer #2: Yes

4. Is the manuscript presented in an intelligible fashion and written in standard English?

Reviewer #1: No

Reviewer #2: Yes

5. Review Comments to the Author

Reviewer #1: 1 The logic of this paper is unclear. It should has been concentrated on the comparison of different protocols and evaluation of alternatives.

2 The statistical analysis is not intuitional.

3 Incorrect subscripts, inconsistent formatting and inappropriate subtitles alongside causing ambiguity.

4 The paper has not given alternative strategies, authors should hold further discussions.

Reviewer #2: Single cell sequencing is getting more popular. In this manuscript, the authors compared the performance of smart-seq, which is deep sequencing and the MARS-seq, the shallower one. Their findings indicated useful hints for choosing the protocol depending on the research purpose. It is also interesting to know that smart-seq generally identified more genes and agrees with the majority of genes identified using the other two methods. The manuscript is well written and is also a good learning material. It is not difficult to understand with the well-presented figures. I only get one question. Gene Cyp8b1 indicated flat curve using smart-seq method, which is different with MARS-seq results. Does it mean that the gene expression scale depends on the method applied? If so, why only one gene act like this, not the other genes?

Generally, I think the manuscript indicated interesting results and shows the privilege of using Smart-seq, I would advise the manuscript to be accepted.

6. PLOS authors have the option to publish the peer review history of their article (what does this mean?). If published, this will include your full peer review and any attached files.

Reviewer #1: No

Reviewer #2: No

---

## [Author Response · Author response to Decision Letter 0]

21 Aug 2020

Response to Reviewers is provided in the uploaded documents.

---

## [Decision Letter · Decision Letter 1]

14 Sep 2020

Reproducibility across single-cell RNA-seq protocols for spatial ordering analysis

PONE-D-20-17601R1

Dear Dr. Bacher,

We’re pleased to inform you that your manuscript has been judged scientifically suitable for publication and will be formally accepted for publication once it meets all outstanding technical requirements.

Kind regards,

Zhong-Hua Chen, Ph.D.

Academic Editor

PLOS ONE

Additional Editor Comments (optional):

Reviewers' comments:

Reviewer's Responses to Questions

**Comments to the Author**

1. If the authors have adequately addressed your comments raised in a previous round of review and you feel that this manuscript is now acceptable for publication, you may indicate that here to bypass the “Comments to the Author” section, enter your conflict of interest statement in the “Confidential to Editor” section, and submit your "Accept" recommendation.

Reviewer #1: All comments have been addressed

Reviewer #2: All comments have been addressed

2. Is the manuscript technically sound, and do the data support the conclusions?

Reviewer #1: Yes

Reviewer #2: Yes

3. Has the statistical analysis been performed appropriately and rigorously? 

Reviewer #1: Yes

Reviewer #2: Yes

4. Have the authors made all data underlying the findings in their manuscript fully available?

Reviewer #1: Yes

Reviewer #2: Yes

5. Is the manuscript presented in an intelligible fashion and written in standard English?

Reviewer #1: Yes

Reviewer #2: Yes

6. Review Comments to the Author

Reviewer #1: (No Response)

Reviewer #2: The authors have addressed the comments properly. I don't think any further comments for this revised version are required.

7. PLOS authors have the option to publish the peer review history of their article (what does this mean?). If published, this will include your full peer review and any attached files.

Reviewer #1: No

Reviewer #2: No

---

## [Editor Report · Acceptance letter]

18 Sep 2020

PONE-D-20-17601R1 

Reproducibility across single-cell RNA-seq protocols for spatial ordering analysis 

Dear Dr. Bacher:

I'm pleased to inform you that your manuscript has been deemed suitable for publication in PLOS ONE. Congratulations! Your manuscript is now with our production department. 

Kind regards, 

on behalf of

Dr. Zhong-Hua Chen 

Academic Editor

PLOS ONE